

# Aging and cardiovascular complexity: effect of the length of RR tachograms

Karthi Balasubramanian[1] and Nithin Nagaraj[2]

[1] Department of Electronics and Communication Engineering, Amrita School of Engineering, Coimbatore, Amrita Vishwa Vidyapeetham, Amrita University, India
[2] Consciousness Studies Programme, National Institute of Advanced Studies, Bengaluru, Karnataka, India

## ABSTRACT

As we age, our hearts undergo changes that result in a reduction in complexity of physiological interactions between different control mechanisms. This results in a potential risk of cardiovascular diseases which are the number one cause of death globally. Since cardiac signals are nonstationary and nonlinear in nature, complexity measures are better suited to handle such data. In this study, three complexity measures are used, namely Lempel–Ziv complexity (LZ), Sample Entropy (SampEn) and Effort-To-Compress (ETC). We determined the minimum length of RR tachogram required for characterizing complexity of healthy young and healthy old hearts. All the three measures indicated significantly lower complexity values for older subjects than younger ones. However, the minimum length of heart-beat interval data needed differs for the three measures, with LZ and ETC needing as low as 10 samples, whereas SampEn requires at least 80 samples. Our study indicates that complexity measures such as LZ and ETC are good candidates for the analysis of cardiovascular dynamics since they are able to work with very short RR tachograms.

# INTRODUCTION

It is well known that functions of all physiological systems are greatly altered during the process of aging. Among these, the cardiovascular system has received prominent attention due to the high death rate attributed to heart related diseases. In fact, the World Health Organization (WHO) has labeled cardiovascular diseases as the number one cause of death globally (*Mendis, Puska & Norrving, 2011*). Hence, cardiac aging has been an important area for research and clinical studies, and understanding heart rate patterns is an essential step in this study.

Heart rate variability (HRV), defined as the variation in the interval between consecutive heart beats, has been proposed as a promising noninvasive quantitative marker for studying autonomic activity, useful for research and clinical studies (*Dreifus et al., 1993*). HRV has been adopted as the conventionally accepted term to describe variations of both instantaneous heart rate and the time interval between successive heart beats. While dealing with consecutive cardiac cycles, researchers have used other terms like heart period

Corresponding author
Karthi Balasubramanian,
b_karthi@cb.amrita.edu

[1]R corresponds to the peak of the QRS complex of the ECG signal and heart beat interval is defined as the time (measured in seconds) between two successive R peaks. RR tachogram is defined as the collection of all the heart beat intervals in the entire ECG signal.

variability, RR variability and RR tachogram[1] to stress that it is the time interval between successive beats and not the heart rate (*TF of the European Society of Cardiology, 1996*). In this paper, we use the term RR tachogram to refer to the beat-to-beat time series.

HRV characterization was initially done by linear methods which included time domain and power spectral density analysis (*Akselrod et al., 1981*; *Schechtman, Kluge & Harper, 1988*). However, it was noticed that the RR tachogram under normal conditions revealed behaviour typical of dynamical systems (*Goldberger, 1991*). This necessitated the use of non-linear dynamics and chaos theory to extract useful information. It was a surprise when further studies led to the discovery that cardiac chaos is not found in pathological systems but instead in the dynamics of normal sinus rhythm (*Goldberger & Rigney, 1990*; *Rössler, Gotz & Rössler, 1979*; *Wessel, Riedl & Kurths, 2009*).

Goldberger, in his seminal work on heart beat dynamics (*Goldberger, 1991*), pointed out that even during bed rest, the heart rate in healthy individuals is not constant or periodic, but shows irregular patterns that are typically associated with a chaotic non-linear system. Spectral analysis on normal heart time series reveals a broad spectrum with a continuum of frequency components, a signature of chaos. Added to this, the phase space mapping of the RR tachogram produced strange attractors, not limit cycles, which clearly indicates the presence of underlying chaotic dynamics in the heart rate variability data. It has been surmised that this chaotic behaviour is due to the fact that a healthy physiological condition is defined by complex interactions between multiple control mechanisms. These interactions are essential for the individual to adapt to the ever changing external environment. These highly adaptive and complex mechanisms lead to a very irregular firing rate of the pacemaker cells in the heart's sinus node, resulting in a chaotic heart rate variability (*Goldberger, 1991*).

The process of aging results in progressive impairment of functional components and alteration of nonlinear coupling amongst them, which in turn impairs the ability of these control mechanisms to maintain a high level of complexity (*Lipsitz & Goldberger, 1992*). Due to these effects, the physiological functions no longer show multitudes of variations but slowly settle down to a more regular nature. This results in reduced nonlinear heart rate fluctuations (*Beckers, Verheyden & Aubert, 2006*) and a progressive loss of complexity in RR interval variability from middle age to old age (*Pikkujämsä et al., 1999*). This has motivated researchers to employ complexity measures (such as approximate entropy, Lempel–Ziv complexity and other information-theoretic and complexity measures) to study cardiac aging, as such measures are known to aid the detection of the regularity inherent in the cardiac time series.

In order to capture this reduction in complexity of cardiovascular dynamics, researchers have employed a variety of techniques such as approximate entropy (*Kaplan et al., 1991*), fractal measures (*Glenny et al., 1991*; *Iyengar et al., 1996*), multivariate embedding methods (*Porta et al., 2014*), band-limited transfer entropy (*Nemati et al., 2013*), multifractality and wavelet analysis (*Humeau et al., 2008*). Most of these algorithms require a minimum amount of data to yield meaningful results and at the same time require that the data be stationary. For physiological data, this may be contradictory as, longer the data set, the more the transitions between different regulatory states, thus giving

rise to nonstationarities. Also, longer measurement data may not be practically available (or reliable) for a number of reasons. Furthermore, long data sets are not able to track fast changes in the respective regulations (*TF of the European Society of Cardiology, 1996*). For HRV analysis, the Task Force of the European Society of Cardiology and the North American Society of Pacing Electrophysiology (*TF of the European Society of Cardiology, 1996*) recommends using only short-term ECG recordings. Hence it is imperative that we use short data lengths for analysis purposes.

Theoretically speaking, use of short data sequences should not pose any problem since heart beat variability time series possess a fractal nature that will enable us to fully characterize the structure with only a limited number of data points. But when it comes to practical implementation, it is difficult since most algorithms require minimum amount of data to effectively analyze and produce acceptable results. For example, information theoretic measures based on Shannon entropy utilize the probability of occurrence of data points which makes it practical only when dealing with huge length of data (*Ebeling, Steuer & Titchener, 2001*).

In this regard, lossless compression algorithm based measures such as Lempel Ziv complexity (LZ) and Effort-To-Compress (ETC) are better in dealing with short sequences (*Amigó et al., 2004*). LZ is a very popular measure used across a wide variety of biomedical applications, including HRV analysis (*Ferrario, Signorini & Cerutti, 2004*). Another measure, Sample entropy (SampEn) which is an improvement over Approximate Entropy measure is used by *Kaplan et al. (1991)* for studying cardiac aging as mentioned earlier. Both LZ and SampEn have been shown effective in in characterizing biomedical signals (*Aboy et al., 2006*; *Aboy et al., 2007*). ETC is a newly proposed measure and is being tested for the first time on cardiac data. However, ETC has been effective in characterizing complexity in an efficient manner for chaotic, stochastic simulated data which are short and noisy (*Nagaraj, Balasubramanian & Dey, 2013*; *Balasubramanian, Nair & Nagaraj, 2015*). Thus ETC is also a good candidate for analysis of short length data.

We are motivated to study the minimum length that is needed for characterizing complexity of cardiac data. To this end, we compare the performance of these measures in their ability to characterize complexity of very short RR tachograms of young and old healthy adults. All the three measures indicate significantly lower complexity values for older subjects than younger ones. We are interested in determining the minimum length of RR tachogram data needed for these measures.

The use of short HRV analysis for applications is increasing since it enables the ambulatory sector to monitor temporary examinations and obtain quick results, hence providing real time monitoring of cardiovascular functions (*Voss et al., 2012*). Some studies that have used short term ECG recordings of 5 min and less (RR tachogram of length of around 300 samples) are given in *Weippert et al. (2014)*, *Sungnoon et al. (2012)*, *Greiser et al. (2009)*, *Nunan, Sandercock & Brodie (2010)*, *Lipsitz (1995)* and *Takahashi et al. (2012)*. *Takahashi et al. (2012)* has used conditional entropy and symbolic analysis with sequences of length 200 samples for differentiating between heart beat intervals of young and old subjects. There have been no studies in literature (to the best of our knowledge) that work with even lesser data.

In the next section, we discuss the experiment setup, followed by a brief introduction to the three complexity measures that we have used in our study and the corresponding statistical analysis performed. Results are presented and analyzed after that and we conclude with discussions and future research directions in the last two sections.

## MATERIALS AND METHODS

### Subjects

Data used for our experiment was obtained from 'Physionet: Fantasia database' (*Goldberger et al., 2000*) and has been described in *Iyengar et al. (1996)*. The experimental details is being succintly summarized here. Two groups of healthy adults: twenty young (age 21–34) and twenty elderly (age 68–81) with 10 males and females in each category participated in the study. All provided written consents and underwent blood count, biochemical analysis and electrocardiogram (ECG) tests. They were healthy, non-smokers with no medical problems and not under any medications and had normal exercise tolerance tests.

### Data acquisition

In *Iyengar et al. (1996)*, the subjects were studied while lying in a supine position and watching the Disney movie-Fantasia to maintain wakefulness. 120 min of continuous ECG data was recorded and digitized by sampling at 250 Hz. The heartbeats were annotated using an automated arrhythmia detection algorithm and the beat annotations were verified by inspecting them visually. The occurrence of each R peak is noted and the time series consisting of the time difference between successive peaks is generated. This process is repeated for each of the participants. This set of time series corresponds to the RR tachograms used in this study.

These RR tachograms have been used for our analysis. We take short length samples from random time instances and analyze them using three complexity measures to gauge their efficiency in distinguishing between the two groups of data.

### Complexity measures

#### *Lempel–Ziv (LZ) complexity*

LZ complexity is a popular measure in the field of biomedical data characterization (*Goez-Pilar et al., 2014*; *Xia et al., 2014*). To compute the LZ complexity, the given data (if numerical) has to be first converted to a symbolic sequence. This symbolic sequence is then parsed from left to right to identify the number of distinct patterns present in the sequence. This method of parsing is proposed in the seminal work on Lempel–Ziv complexity (*Lempel & Ziv, 1976*). The very succinct description in *Hu, Gao & Principe (2006)* is reproduced here.

Let $S = s_1 s_2 \cdots s_n$ denote a symbolic sequence; $S(i,j)$ denote a substring of $S$ that starts at position $i$ and ends at position $j$; $V(S)$ denote the set of all substrings ($S(i,j)$ for $i = 1, 2, \ldots n$; and $j \geq i$). For example, let $S = abc$, then $V(S) = \{a, b, c, ab, bc, abc\}$. The parsing mechanism involves a left-to-right scan of the symbolic sequence $S$. Start with $i = 1$ and $j = 1$. A substring $S(i,j)$ is compared with all strings in $V(S(i, j-1))$ (Let $V(S(1,0))$ $= \{\}$, the empty set). If $S(i,j)$ is present in $V(S(1, j-1))$, then increase $j$ by 1 and repeat

the process. If the substring is not present, then place a dot after $S(i,j)$ to indicate the end of a new component, set $i = j + 1$, increase $j$ by 1, and the process continues. This parsing procedure continues until $j = n$, where $n$ is the length of the symbolic sequence. For example, the sequence '*aacgacga*' is parsed as '*a.ac.g.acga.*'. By convention, a dot is placed after the last element of the symbolic sequence and the number of dots gives us the number of distinct words which is taken as the LZ complexity, denoted by $c(n)$. In this example, the number of distinct words (LZ complexity) is 4. Since we may need to compare sequences of different lengths, a normalized measure is proposed and is denoted by $C_{LZ}$ and expressed as:

$$C_{LZ} = (c(n)/n)\log_\alpha n. \tag{1}$$

where $\alpha$ denotes the number of unique symbols in the symbol set (*Aboy et al., 2006*).

### Sample entropy (SampEn)

Sample entropy (*Richman & Moorman, 2000*) is a complexity measure used to quantify regularity of time series, especially short and noisy sequences. It is a measure that monitors how much a set of patterns that are close together for a few observations, still retain its closeness on comparing the next few observations. Sample entropy has its roots in another closely related measure, approximate entropy (ApEn) that was proposed by *Pincus (1995)*. ApEn was used in different cardiovascular related studies including (*Kaplan et al., 1991*; *Acharya et al., 2006*; *Al-Angari & Sahakian, 2007*). But it suffers from drawbacks that include: bias towards regularity, dependency on data length and lack of consistency with respect to parameter changes (*Richman & Moorman, 2000*; *Yentes et al., 2013*). Sample entropy doesn't suffer from these and hence is a better choice as a complexity measure. Two input parameters, $m$ and $r$ must be initially chosen for the computation of the measure—$m$ being the length of the patterns we want to compare each time for closeness and $r$, a tolerance factor for the regularity of the two sets of patterns being compared.

### Effort-To-Compress (ETC) complexity

Effort-To-Compress (ETC) is a recently proposed complexity measure that is based on the effort required by a lossless compression algorithm to compress a given sequence (*Nagaraj, Balasubramanian & Dey, 2013*). Similar to the computation of LZ complexity measure, the given data (if numerical) has to be first converted to a symbolic sequence. At each iteration, the algorithm replaces the most frequently occurring pair of symbols with a new symbol resulting in a transformed sequence. This is repeated until the transformed sequence is a constant sequence. For example, the input sequence '11010010' is transformed into '12202' since the pair '10' has maximum number of occurrences compared to other pairs ('00', '01' and '11'). In the second iteration, '12202' is transformed to '3202' since '12' has maximum frequency (in fact all pairs are equally likely). The algorithm proceeds in this fashion until the length of the string is 1 or the string becomes a constant sequence (at which stage the entropy is zero and the algorithm halts). In this example, the algorithm transforms the input sequence '11010010' $\mapsto$ '12202' $\mapsto$ '3202' $\mapsto$ '402' $\mapsto$ '52' $\mapsto$ '6'.

The ETC measure is defined as $N$, the number of iterations required for the input sequence of length $L$ to be transformed to a constant sequence. In the above example $N = 5$. The normalized version of the measure is given by: $\frac{N}{L-1}$ (Note: $0 \leq \frac{N}{L-1} \leq 1$).

## Statistical analysis

In our research, we are interested in finding out the minimum length of RR tachograms required for effectively discriminating between the two groups of subjects: young and old healthy adults. The analysis procedure is as follows:

### Analysis procedure

- From each of the twenty young and old data sets, choose consecutive $L$ number of samples from a random location.
- Calculate the SampEn, LZ and ETC complexity measures for the chosen $L$ length data set.
- For statistical accuracy, 50 such locations are randomly chosen and 50 complexity values for each of the measures are calculated.
- The complexity assigned to each of the measure for a sequence of length $L$ is the average of all the 50 values calculated.
- Thus for each complexity measure, we obtain 20 complexity values for the young subjects and 20 complexity values for the old subjects.
- Using these values as samples representing the young and old populations, a two-sample $t$-test is performed for each complexity measure.
- The results of the $t$-test are analyzed to check if the mean complexity value of the RR tachograms of the young subjects is significantly greater than the mean complexity value of the RR tachograms of the old subjects.
- A non-parametric test (Mann–Whitney $U$-test) is also performed to further validate the results.
- The entire process is repeated multiple times with different values of $L$ (varying from 8 to 500) and the minimum length at which each complexity measure is able to successfully differentiate, is determined.

### Sequence length considerations

As per the procedure outlined above, complexity measures were calculated for data of different lengths. Since approximate entropy and sample entropy were proposed with the intention of handling short and noisy sequences, there has been some concerted efforts by researchers in using these measures to characterize short length data sequences. *Yentes et al. (2013)* point out that sample entropy works well with data lengths of 200 samples or more. *Abásolo et al. (2006a)* suggest that a data length of $10^m - 20^m$ is required to calculate sample entropy of EEG signals.[2] *Yang et al. (2013)* also point out in their study of fMRI analysis that a data of $10^m - 20^m$ samples is required. This translates to data lengths greater than 100 samples. However *Sokunbi (2014)* have shown that 85 samples are enough for sample entropy to effectively discriminate fMRI data of young and old adults. In our experiment, we chose to analyse with lesser data and found that 80 samples are enough for sample

[2] $m$ for sample entropy calculations is generally taken as 2 or 3.

[3]While analyzing with less than 80 samples, sample entropy diverged to infinity and didn't give meaningful values.

entropy ($m = 2$ and $r = 0.2SD$) to effectively distinguish between RR tachograms of young and old adults.[3]

### Parameter settings

For calculation of sample entropy, values of $r$ and $m$ need to be fixed and for ETC and LZ complexities, the number of bins has to be decided. It is to be noted that, for LZ and ETC, the continuous valued RR tachograms were quantized using 4 bins and again using 8 bins and complexity measures were calculated for sequences of different lengths. We first look at the reason behind the choice of these parameters before presenting the results.

*Sample entropy: choice of $m$ and $r$.* In the calculation of sample entropy, the values of the parameters $m$ and $r$ needs to be chosen carefully. There has been no consensus established to select the parameters. Researchers in general have suggested using $r$ value between 10% to 25% of the *SD* (standard deviation) of the input data and $m$ of 1 or 2. They have also recommended that a range of $r$ and $m$ values should be tested for the particular data set under consideration before fixing a value for analysis (*Pincus, 1995*; *Pincus & Goldberger, 1994*; *Liu et al., 2010*; *Mayer et al., 2014*; *Yentes et al., 2013*; *Groome et al., 1999*).

We experimented with multiple values and found that small values of $r$ (less than 0.2) or high values of $m$ (3 and more) made the sample entropy value to diverge to infinity. By choosing $m = 2$ and $r = 0.2SD$ or $0.3SD$, sample entropy could differentiate between younger and elderly subjects and also the complexity value was higher for the younger subjects as compared to the older subjects, as is expected. At the same time, by choosing $m = 2$ and $r = 0.25SD$, sample entropy was able to differentiate between the two groups, but the elderly adults displayed higher complexity, which was not as expected. Thus, we propose that $m = 2$ and $r = 0.2SD$ or $0.3SD$ are good choices for discriminating between the RR tachograms of young and elderly adults. It is advisable to not use $r = 0.25SD$ since it incorrectly assigns a higher complexity value to elderly subjects as compared to younger subjects.

*ETC and LZ: how to decide on the number of bins?* LZ and ETC complexity measures are based on coarse-graining of the measurements where the continuous valued time series need to be converted into discrete symbolic sequences before computation of the complexity measures. The range of sampled values is first divided in to sections (bins) of uniform width and the sampled values are then assigned to one of the bins, thus discretizing the continuous values.

Quantization, or binning the given continuous data in to discrete intervals is a data fitting process where the model is defined by the number of bins and the size of the bin intervals. In order to model with least error, bias–variance trade-off needs to be considered. The bias error, defined as the difference between the predicted (binned) and the actual values is large when the number of bins used is very small. On the other hand, when we increase the number of bins, the model gets over-fitted. This causes severe errors, especially in the case of noisy systems where noise is captured as data variations by the over-fitted model. Researchers using LZ complexity have used both 2 bins (for e.g., *Abásolo et al. (2006b)*) as well as 4 bins (for e.g., *Kamath (2013)*). In our study of heart rate analysis,

**Table 1  Student's *t*-test and Mann–Whitney *U*-test results for SampEn (*m* = 0.2) to distinguish between beat-to-beat intervals of healthy young and old subjects (*df* refers to degree of freedom).**

| Length | r (SD) | Mean ± SD (Old) | Mean ± SD (Young) | *t*-value | *df* | *p* | Mann–Whitney *U*-value | Significant difference |
|---|---|---|---|---|---|---|---|---|
| *L* = 100 | 0.2 | 1.134 ± 0.475 | 1.515 ± 0.428 | −2.66 | 37 | 0.006 | 113 | yes |
| | 0.3 | 0.862 ± 0.395 | 1.120 ± 0.389 | −2.08 | 37 | 0.022 | 125 | yes |
| *L* = 80 | 0.2 | 1.146 ± 0.460 | 1.514 ± 0.443 | −2.58 | 37 | 0.007 | 111 | yes |
| | 0.3 | 0.861 ± 0.403 | 1.122 ± 0.401 | −2.05 | 37 | 0.024 | 124 | yes |

**Table 2  Student's *t*-test and Mann–Whitney *U*-test results for LZ and ETC complexity analysis to distinguish between beat-to-beat intervals of healthy young and old subjects using four bins (*df* refers to degree of freedom).**

| Length | Complexity | Mean ± SD (Old) | Mean ± SD (Young) | *t*-value | *df* | *p* | Mann–Whitney *U*-value | Significant difference |
|---|---|---|---|---|---|---|---|---|
| *L* = 20 | LZ | 0.871 ± 0.079 | 0.928 ± 0.047 | −2.76 | 30 | 0.005 | 115 | yes |
| | ETC | 0.712 ± 0.035 | 0.729 ± 0.018 | −1.86 | 28 | 0.037 | 134 | yes |
| *L* = 15 | LZ | 0.905 ± 0.067 | 0.948 ± 0.049 | −2.36 | 34 | 0.012 | 119 | yes |
| | ETC | 0.771 ± 0.032 | 0.778 ± 0.015 | −0.87 | 27 | 0.195 | 197 | no |
| *L* = 10 | LZ | 0.958 ± 0.048 | 0.963 ± 0.048 | −0.38 | 38 | 0.352 | 183 | no |
| | ETC | 0.869 ± 0.037 | 0.871 ± 0.031 | −0.15 | 38 | 0.439 | 198 | no |

we experimented with different bins, namely two, four, eight and 16 and found that the characterization was poor using two and 16 bins but was satisfactory with four and eight bins. Hence we decided to use four and eight bins for our analysis. It is to be noted that the optimum bin size may vary from application to application and hence it is advisable to test results with multiple bins for each of the measures being used.

## RESULTS

Two-sample *t*-test (assuming unequal variances) was performed to identify if there are significant differences in the complexity values of RR tachograms of young and old adults. For the sake of being conservative, we have also performed the Mann–Whitney *U*-test and the *U*-values are reported.[4] Table 1 shows the results for sample entropy while Tables 2 and 3 give the results for the ETC and LZ complexity measures.

The *t*-test results for analysis may be summarized as follows:

- The mean SampEn complexity of the RR tachograms of old subjects is significantly less than that of young subjects for data lengths of 80 and higher.
- For analysis with four bins, the mean LZ complexity of RR tachograms of old subjects is significantly less than that of young subjects for data lengths of 15 and higher while the same is true for data lengths of 10 and higher, while using eight bins.
- For analysis with four bins, the mean ETC complexity of the RR tachograms of old subjects is significantly less than that of young subjects for data lengths of 20 and higher while the same is true for data lengths of 10 and higher, while using eight bins.

[4]The critical value of *U* at *p* < 0.05 is 138.

**Table 3  Student's *t*-test and Mann–Whitney *U*-test results for LZ and ETC complexity analysis to distinguish between beat-to-beat intervals of healthy young and old subjects using eight bins (*df* refers to degree of freedom).**

| Length | Complexity | Mean ± SD (Old) | Mean ± SD (Young) | *t*-value | *df* | *p* | Mann–Whitney *U*-value | Significant difference |
|---|---|---|---|---|---|---|---|---|
| L = 15 | LZ | 0.795 ± 0.045 | 0.832 ± 0.023 | −3.09 | 32 | 0.002 | 83 | yes |
| | ETC | 0.883 ± 0.033 | 0.909 ± 0.018 | −3.01 | 29 | 0.003 | 78 | yes |
| L = 10 | LZ | 0.788 ± 0.041 | 0.809 ± 0.021 | −2.10 | 28 | 0.023 | 134 | yes |
| | ETC | 0.933 ± 0.030 | 0.950 ± 0.013 | −2.30 | 25 | 0.015 | 119 | yes |
| L = 8 | LZ | 0.775 ± 0.033 | 0.790 ± 0.023 | −1.55 | 33 | 0.066 | 160 | no |
| | ETC | 0.956 ± 0.019 | 0.962 ± 0.013 | −1.25 | 34 | 0.109 | 161 | no |

Further, nonparametric Mann–Whitney U tests were also done and the interpretations from the data do not change.

## DISCUSSION

This study presents the first rigorous examination of the effect of the length of RR tachograms on the ability of complexity measures to segregate HRV data as belonging to either young or old healthy subjects. The results of our study indicate that even with shorter lengths of RR tachograms, there is a statistically significant difference between the complexities of young and old healthy adult hearts. This might reflect some form of age related physiological changes suggesting susceptibility to disease.

There are important differences related to the length at which this difference is significant for the three measures. Sample entropy needs a minimum of 80 samples of the RR tachogram to give finite values for the complexity. Both Lempel–Ziv and ETC show significant differences in complexities for young and old healthy adult hearts for as low as 10 and 15 samples of the RR tachogram. As mentioned before, each sample in the RR tachogram represents the time duration between successive R peaks. An exact temporal value corresponding to these samples can't be made since the time between R peaks is not a constant value. Assuming an average time of 1 sec between successive peaks, we can estimate that 10–15 samples correspond to approximately 10–15 s of ECG data.

Of particular interest is the results obtained by the new complexity measure ETC. As previously mentioned, this is the first time that ETC has been applied for characterizing complexity of RR tachogram. What is encouraging is not only the ability of ETC to show significant statistical differences between the two groups (young and old) for short length of RR tachograms, but also the fact that it records low values of standard deviation. In fact, in all cases, ETC has the least standard deviation among the three measures. Such precise characterization of complexity and separation between the two groups makes ETC a potential candidate for further analysis and for cardiac aging detection and classification.

Studies have observed that a strategy of using several complexity measures in conjunction improves efficiency of classification, owing to the fact that these measures are not very well correlated with each other, indicating that each measure captures a different notion of complexity (*Acharya et al., 2013*; *Burns & Rajan, 2015*). Thus, ETC could be added to the list of existing complexity measures as a promising candidate for combining with other

**Table 4  RR tachogram lengths used in existing studies on HRV.**

| Existing literature | Length of ECG recording | Estimated length of tachogram used (samples) | Complexity measures used |
|---|---|---|---|
| *Pikkujämsä et al. (1999)* | 24 hrs | 86,500 | ApEn, LZ, SampEn |
| *Ferrario, Signorini & Cerutti (2004)* | 24 hrs | 86,500 | LZ |
| *Voss et al. (2012)* | 5 min | 300 | Symbolic dynamics |
| *Weippert et al. (2014)* | 5 min | 300 | SampEn |
| *Sungnoon et al. (2012)* | 5 min | 300 | LZ |
| *Takahashi et al. (2012)* | 15 min | 200[a] | Conditional entropy |

**Notes.**
[a] indicates that the authors had used this specific length of the tachogram.

measures to aid cardiac aging detection and classification. Determining the exact reasons for the lower precision of ETC for short time series is an ongoing research effort.

## Comparison with other approaches

As previously noted, researchers studying cardiac aging have used a number of complexity measures. Table 4 shows a sample of studies used in literature (not exhaustive), complexity measures they have used and the length of RR tachograms in their respective studies. In some of these studies, the actual length of RR tachogram used was known and is reported in the table. However, in others, we did not know the exact length but have estimated it by assuming that each RR interval on an average would be approximately one second duration long.

In several studies, LZ and SampEn complexity measures are preferred over a number of other information theoretic measures. One of the reasons as cited before is that these measures perform well, even with as low as 200–300 samples of RR tachograms. Our study is very relevant in this context as we are interested in studying the minimum length of RR tachograms needed for these measures to discriminate between the two groups (young and old healthy hearts).

Figure 1 shows the effect of length of RR tachogram on the complexity values (on young and old) for the three measures—SampEn, LZ and ETC respectively. We have plotted the means of the complexity value for various tachogram lengths.

Some observations worth noting are:

- All three complexity measures predict higher complexity for the younger subjects as compared to the elderly, for all the lengths considered. This shows that relative consistency is maintained and the measures are able to distinguish between the two groups even though the complexity measures, by themselves are changing with length.
- LZ and ETC show a downward trend for the mean complexity value for both the groups (young and old) whereas SampEn shows roughly a constant trend as the length of the RR tachogram is varied from forty[5] to five hundred samples. It is known from previous work (*Amigó et al. (2004)* and yet to be published work) that LZ and ETC yield constant complexity measures as the length of data increases, only for stationary Markov processes. Since ECG signals are nonstationary, we don't see such a behaviour for LZ

[5] Please note that sample entropy showed meaningful values only for lengths greater than 80 samples.

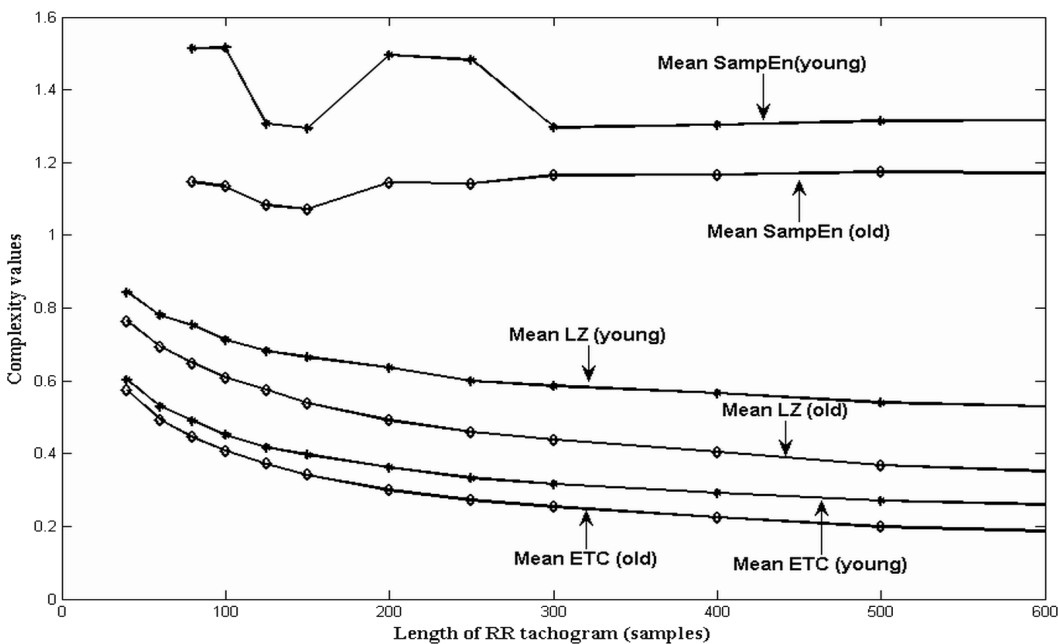

**Figure 1 Change in SampEn, LZ and ETC of HRV of young and old subjects with varying lengths of RR tachogram.**

and ETC. The fact that the means are decreasing for both LZ and ETC indicates that there is some regularity in the RR tachograms which is revealed as the length increases. A completely random sequence of the same length will have a much higher value of complexity as compared to that of the RR tachogram.

- As length of RR tachogram increases, we find that the standard deviation increases for all the three measures. However, ETC has the best precision as indicated by a lower value of standard deviation which is also reflected in the plots. This is a very desirable feature.
- In order to measure the dispersion of the three measures, we compute the coefficient of variation (CoV) defined as the ratio of standard deviation to the mean ($\frac{\sigma}{\mu}$) and plot the same in Figs. 2 and 3, for young and old healthy heart data respectively. ETC has the best (least) value of coefficient of variation for all lengths of the RR tachogram. The CoV for LZ and ETC worsens (increases) with increasing length of RR tachogram whereas it remains roughly constant for SampEn.

Based on these plots and the observations, we conclude that at large lengths, it is preferable to choose SampEn over LZ and ETC, and at smaller lenghts, ETC is to be preferred over the other two. In any case, it is better to use more than one measure in conjunction (see *Acharya et al. (2013)* and *Burns & Rajan (2015)*) and ETC is definitely a promising candidate given its low value of CoV, especially for short RR tachograms.

## Utility of our approach

Introducing new measures for analysis of cardiac data is desirable since it may throw new insights into the underlying dynamics. ETC is a new measure which the authors had
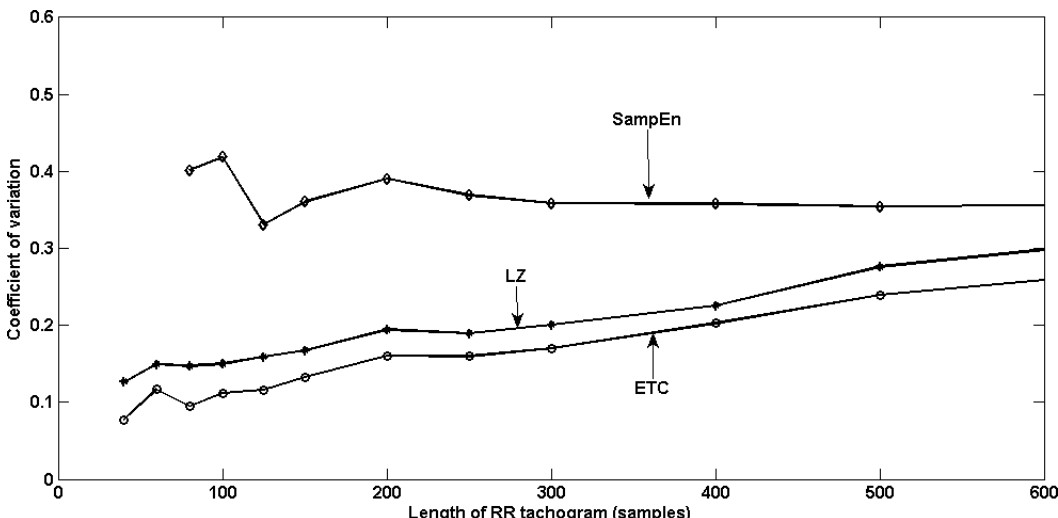

**Figure 2  Coefficient of variation of HRV of old subjects with varying lengths of RR tachogram.**

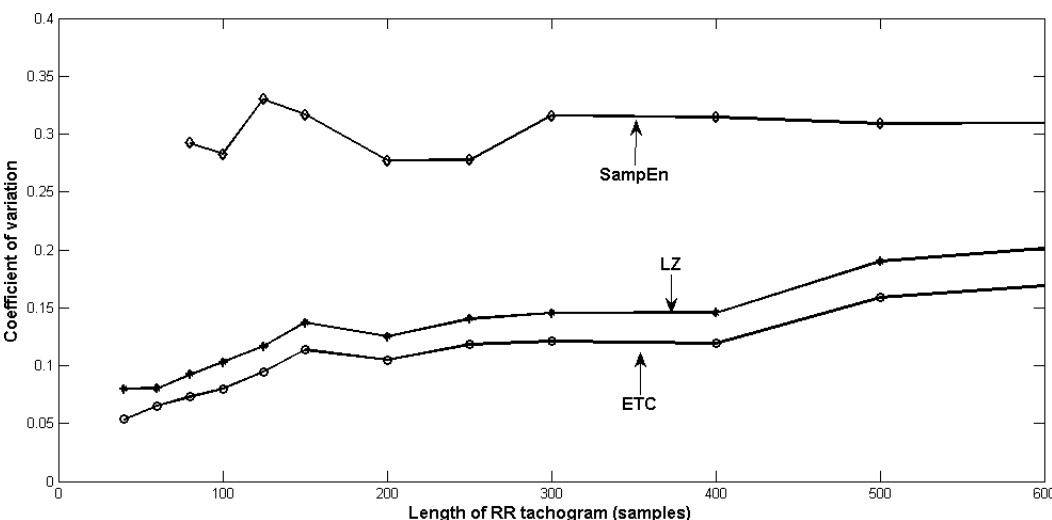

**Figure 3  Coefficient of variation of HRV of young subjects with varying lengths of RR tachogram.**

proposed in the context of time series analysis and classification. Unlike Shannon entropy and Lempel–Ziv complexity, both of which characterize the degree of compressibility, ETC measures the "Effort-To-Compress" the input sequence (after the original data is converted into a sequence of symbols by binning). In many cases, the degree of compressibility may not reflect the actual complexity of the data. For e.g., consider two sequences $A = 101010$ and $B = 110001$:

- Both have the same number of zeros and ones, thus possessing the same first-order empirically estimated entropy value (entropy of $A =$ entropy of $B = 1$).
- LZ (Lempel Ziv) complexity of $A$ (1.0.101.0.) = 4 and LZ (Lempel Ziv) complexity of $B$ (1.10.00.1.) = 4.

It is evident that sequence $A$, being periodic, is less complex than sequence $B$, but entropy and LZ complexity measures are unable to distinguish between the two sequences. On the other hand, the ETC measure of $A = 1$ ('101010' $\mapsto$ '222') and of $B = 5$ ('110001' $\mapsto$ '20001' $\mapsto$ '2301' $\mapsto$ '401' $\mapsto$ '51' $\mapsto$ '6'), reflecting the ability of ETC to capture the randomness present in the sequence and distinguish between the two. ETC is thus capturing a different feature of the data which is not captured by the existing measures. Since ETC of the RR tachogram for healthy young adults is higher (statistically) than that of healthy older adults, it indicates that the effort to compress RR tachogram of a younger heart is higher than that of older heart. We don't yet fully understand the implications of this observation, but it seems to suggest that the inherent RR tachogram complexity of a young healthy heart is of a fundamental nature. If this were not the case, ETC should have recorded lower values of complexity. Further explorations are needed to make more conclusive inferences.

### Limitations of the study

This study involved healthy young and old subjects and the difference in complexity values might not be related to any pathological changes but only to some form of physiological changes that are embedded in the entire recording. Hence, the study doesn't indicate the usefulness of the complexity measures to detect pathological changes where changes appear to switch abruptly between normal and abnormal stages. It is worthwhile to repeat similar analysis in the future, but with pathological findings like arrhythmia and other heart related abnormalities.

## CONCLUSIONS AND FUTURE RESEARCH WORK

As noted previously, complexity measures are very popular for studying cardiac data and they are increasingly being used for detection, analysis and classification in cardiac applications. It is imperative that we understand the limits of these measures, especially with respect to length of data. Our study, for the first time, has empirically determined the lower limit for the length of RR tachogram data to discriminate between two groups - namely young and old healthy adult hearts. We have considered three measures, two of which (Sample Entropy and Lempel–Ziv Complexity) are popularly used in several previous studies, whereas the third measure (ETC) is used for the first time.

Based on our study on the experimental data, at a 5% significance level (overall error rate) for the statistical test, there is sufficient evidence to conclude that:

- SampEn is able to distinguish between RR tachograms of young and old adults for data lengths of 80 and more.
- For data analyzed using four bins, LZ complexity measure is able to distinguish between RR tachograms of young and old subjects for lengths of 15 or more, while ETC complexity measure is able to do so for lengths of 20 and higher.
- For data analyzed using eight bins, both LZ and ETC complexity measures are able to distinguish between RR tachograms of young and old subjects for lengths as short as 10 data samples.

The lower coefficient of variation of ETC, makes it a promising candidate for further investigation in cardiac applications involving detection, analysis and classification.

For future research work, it is imperative that we study the effect of noise (which is unavoidable in real-life applications) and missing-data on the complexity measures and how it might impact the discrimination between cardiac signals of young and old subjects. This could be done by adding different levels of noise on the existing data-sets and also removing parts of data to simulate missing-data problem. One area of research which we have not focused in the current study is the effect of bin structure on the performance of these complexity measures (LZ and ETC). We have chosen four and eight bins of uniform size in our study. It would be interesting to see what happens to the performance of these measures when the number of bins, as well as the size of the bins (non-uniform and adaptive) are changed. This is a direction which we wish to undertake as a part of our future research.

Given the recommendation of the Task Force of the European society of Cardiology and the North American Society of Pacing Electrophysiology to use short-term ECG recordings for HRV analysis (*TF of the European Society of Cardiology, 1996*), it is encouraging to learn from our study that complexity measures are up to this challenge of working with short RR tachogram data and this has potentially significant clinical implications. LZ and ETC measures show significant differences in complexities (for young and old healthy adult) even for small lengths of RR tachograms (as low as 10–15 samples).

## ACKNOWLEDGEMENTS

The authors would like to acknowledge the help rendered by Gayathri R Prabhu (Indian institute of Technology, Chennai), Sriram Devanathan (Amrita University), Del Marshall (Amrita University) and Sandipan Pati (University of Alabama).

### Funding

The authors received no funding for this work.

### Competing Interests
The authors declare there are no competing interests

### Author Contributions
- Karthi Balasubramanian and Nithin Nagaraj conceived and designed the experiments, performed the experiments, analyzed the data, contributed reagents/materials/analysis tools, wrote the paper, prepared figures and/or tables, reviewed drafts of the paper.

### Data Availability
Data for analysis was retrieved from https://physionet.org/physiobank/database/fantasia/.

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
