# Peer review of "Aging and cardiovascular complexity: effect of the length of RR tachograms"

_PeerJ, doi:10.7717/peerj.2755_

## Round 0.1 · original submission · Major Revisions

As you can see, the three reviewers have similar concerns, particularly about the presentation. Please address these concerns clarifying the presentation of the method that you use and the results, and trying to improve the English language, if possible through the help of a native speaker. The reviewers also suggest several articles that you could cite to better put your work in the context of previous research. Please cite the most relevant papers (you do not have to cite all of those that have been suggested) and reduce your reference list eliminating less relevant references as suggested by reviewer 1. Please also address the concern of reviewer 3 about multiple tests.

Reviewer 1 ·

Basic reporting

Authors present a work assessing the effect of the length of the tachogram on complexity indexes. In particular, they found the minimum length of the RR series for discriminating old and young subjects according to Sample Entropy, Lempel Ziv Complexity and Effort-to-compress measures.

The work is clear and easy to be read. Anyway it can be improved in some parts to help researchers to exploit this work in the future.

There are several works studying cardiovascular variability complexity changes with aging, but only few are cited in Introduction. Please add some refs, as [PLoS One. 2014 Feb 24;9(2):e89463; Proc Natl Acad Sci 2002; 99: 2466–2473] and comment on them.
The number of references is elevated, some of them not directly related to the topic might be avoided or replaced with more relevant ones. Furthermore congresses proceedings while not completely attinent can be avoided or replaced with a full paper while existing.

Check the work for mispelling, references as well: some capital letters are missing, the name of EMBS conference is misspelled, lacks of authors and so on.

Figures: Figs2 and 3 could be unified in a unique figure with two panels.
Try to use patterns/colors to differentiate results instead of using arrows and labels in the figure.
Describe the findings obtained by the figures in results (from L233).

Experimental design

NOVELTY
Dataset: the fantasia database is not new to me and I guess previous works investigated complexity on this database, Could you find something in literature, if exists, and comment on the results? Comparing with that would strengthen your hypothesis and highlight the novelty of your findings.

I also have some concerns about METHODS DESCRIPTION:
RR Series extraction: how did you perform it? Did you correct series? According to which criterion? That is necessary to avoid probelms as flattening during binning.

Complexity indexes:
P6 L100 Better explain the LZ complexity.
Sample entropy: give the formula
ETC: the description is not very clear, e.g. at L122: Why new sequence start with 3? Please, better describe the method.

Analysis, L132.
The procedure to define the optimum length L could be improved: what was the minimum and the maximum L tested? I would suggest to give these values for SampEn, LZC, ETC and discuss it depending also on the other parameters (see below).

What were the parameters (m, r …) used for each complexity index? Give them in the text and omit footnotes.
State in the text also that Sample Entropy converged to infinity for L<80.

The choice of the number of bins should be included here. Why did you use 4 and 8, since you stated that in previous works the choice was 2 and 4?(L187). Did anyone use 8 before you?
Consider that the dynamic that can be represented depends on the number of bins but also the number of symbols, so this point should be discussed better.

L153-155 please give some references that investigate cardiovascular series, and not fMRI, EEGs..., with different lengths, as [PLoS One. 2014 Feb 24;9(2):e89463; PLoS One. 2014 Apr 4;9(4):e93808.] Some of them could be added in table IV as well.

Describe in methods how the coefficient of variation given in figs 2 and 3 was obtained.

Validity of the findings

Findings could be better described
Results L190: please better describe tables in the text.

Table1: is it correct m=0.2 in the caption?
Describe in captions all the parameters given in tables. What is "df" referred to?

Additional comments

Discussion: what do you think about the fact that 80 is the shortest length of tachogram for obtaining a concrete value of SampEn but it is also the smallest value able to separate groups?
What could be the differences investigating other groups? It must be reminded that, although the power of complexity measures is high, not every groups can be separated by such indices. Could be 80 the “optimum” length for every population?

Any comment about the comparison of the three methods is welcome. Do you think there is a better one? What are their pros and cons?

Reviewer 2 ·

Basic reporting

No comments

Experimental design

No comments

Validity of the findings

No comments

Additional comments

It is a very good work. I have few concerns. They are indicated below:
(i) Written English need to improved.
(ii) Authors have used SampEn and LZ complexity. They need explain clearly, why they have used these parameters. The focus is missing.
(iii) In my opinion, authors need to highlight the novelty of their work.
(iv) Please include latest papers related to entropies, and nonlinear parameters. Few are given below:
• Computer-based analysis of cardiac state using entropies, recurrence plots and Poincare geometry
• Linear and non-linear analysis of cardiac health in diabetic subjects
• Current methods in electrocardiogram characterization
• Analysis of cardiac signals using spatial filling index and time-frequency domain
• Study of heart rate variability signals at sitting and lying postures
• Application of entropies for automated diagnosis of epilepsy using EEG signals: a review.
(v) Please highlight the advantages and disadvantages of this method.
(vi) Conclusion: Please provide clinical implications.
(vii) Also, please provide the future work
(viii) Authors need to explain Fig.1 and 2 more clearly.
(ix) Authors need to explain how their work id different from papers indicated in Table IV.

Reviewer 3 ·

Basic reporting

In this report the authors present the effect of length of RR tachograms in complexity estimators. The study was aimed to investigate entropy and information compressors suitable to describe the complexity of short EKG time series. They found that LZ and ETC are more efficient to detect differences between RR tachograms from young and old participants when the time series are short. However, for longer time series, SampEn was found to be more adequate because it gave more stable estimations (lower coefficient of variation for SampEn) than LZ and ETC.

The structure of the work is simple, clear and straightforward, and the methodology to obtain the results is also adequate.
In general, figures and tables are easy to interpret and well labeled. Nonetheless, I would have included a figure (or a short explanation) about the structure of RR tachograms. This can be especially useful for experts in other areas of research that are not closely related with EKG.
Language of the manuscript: Although I am not a native English speaker I believe that the language is easy to understand. There are, however, some expressions and errors that I would like to comment and authors may consider for correction:
- Page 1 line 23 to 24: “To describe…”. To me, this sentence is weird from a syntactic point of view.
- Page 1, line 40-41: “Aging results…”. Difficult to understand.
- Page 1, line 31: points should be pointed
- Page 2 line 81: “…has shown promise to characterize…” ¿replace with “…has shown to be a promising…”?
- Page 2, line 98: replace have beeb with have been
- Lempel-Ziv should be replaced with LZ after the first indication of the acronym in the text.

Experimental design

The design and methodology applied in this study is simple and, in general, adequate.
Tables are helpful to understand the significance of the results. Regarding the information provided in the tables, I have two questions for the authors:
In the tables (I, II and III), It can be seen that the degrees of freedom are different depending on the condition and measure. How did the authors get this? Did you use dissimilar sample size for each condition and measure? This should be included in the results section. In principle, if I am not wrong, t-tests for two groups of 20 participants give df = 38 for all conditions and measures. (In the tables provided here you can see that df is in a range: 27 < df < 38).
The other question is about the p-value corrections for multiple comparisons. Given that you needed to apply many t-tests on each length, I believe that it would be nice to correct for false positive comparisons (using False discovery rate, FDR, for example. Bonferroni would be the most conservative). Did you apply any correction in your t-test procedure?

Validity of the findings

Although I believe that these results can be replicated with series from other conditions (patients with heart pathologies for example), It would have been interesting to include additional experimental conditions to be able to extend the validity of the results.

The last concern I would like to point out is related with the relationship between the sampling of the time series and its structure. In the Conclussions section, it is stated that LZ estimations from 10 data samples can give reliable results. My question is related with the equivalent time length of this segment. ¿How many milliseconds do you have in a segment of 10 points? Is it fixed or it is related to R to R appearance? I am not an expert on this field and I do not know how tachograms are constructed, but I understand that from a signal sampled at 250 HZ you obtain a tachogram with much less time resolution. This would explain why only 10 points would be enough for ETC. In any case as I stated before I believe that a short explanation on how these tachograms are obtained would help to expand the scope of potential readers (introduce the explanation in page 2 line 97).

Minor comments:
Page 2, line 106: indicate the meaning of n and C(n) in the formula number 1.
There are many descriptions of results in the discussion I would consider to move these descriptions to the results section. For example, in page 6 lines 239-252 there is a description of the results but not a discussion. It is not stated in the results that the coefficient of variation was calculated and readers might feel a bit confused when they go through this paragraph.

Additional comments

In general, I believe that the manuscript is worthy to be published, but it needs changes as indicated in the other sections of the review

---

## Round 0.2 · Minor Revisions

The paper will be accepted provided the authors address the minor points raised by reviewer 1.

Reviewer 1 ·

Basic reporting

no comment

Experimental design

no comment

Validity of the findings

no comment

Additional comments

The paper has been improved after first revision and concerns have been addressed by authors.
Anyway, before publication, I would suggest to clarify some more few points:
i) at Line154 authors give a minimum L equal to 20, while from tables and Results it appears that at least a minimum length L=8 has been used. Could you please clarify? Was the minimum lenght L tested the same for all measures?
ii) Line253: "all three measures predict lesser complexity for younger.." . Probably that was a mistake and authors meant LARGER instead of LESSER?
iii) Although the language has been improved some typos is left. I would suggest to revise one more time the paper (e.g. Line290 "..an younger...")

Reviewer 2 ·

Basic reporting

Excellent

Experimental design

It looks perfect

Validity of the findings

Very good.

Additional comments

Authors have addressed all my queries satisfactorily. I propose to accept the paper in its present form. Thank you.

Reviewer 3 ·

Basic reporting

I see that the authors have made an effort to make clear some points that might have been difficult to understand for non expert readers.

In addition, the language have been improved, increasing the readability of the manuscript.

Experimental design

No comments

Validity of the findings

No comments

Additional comments

Because all points were addressed and discussed adequately. I recommend this report for publication in PeerJ

---

## Round 0.3 · accepted · Accept

After the last corrections, your paper is now ready for publication.